# The SSTeP-KiZ System—Secure Real-Time Communication Based on Open Web Standards for Multimodal Sensor-Assisted Tele-Psychotherapy

**DOI:** 10.3390/s22249589

**Published:** 2022-12-07

**Authors:** Jonas Primbs, Winfried Ilg, Annika Thierfelder, Björn Severitt, Carolin Sarah Hohnecker, Annika Kristin Alt, Anja Pascher, Ursula Wörz, Heinrich Lautenbacher, Karsten Hollmann, Gottfried Maria Barth, Tobias Renner, Michael Menth

**Affiliations:** 1Department of Computer Science, University of Tübingen, 72076 Tübingen, Germany; 2Section for Computational Sensomotorics, Department of Cognitive Neurology, University of Tübingen, 72076 Tübingen, Germany; 3Child and Adolescent Psychiatry, University Medical Center of Tübingen, 72076 Tübingen, Germany; 4Business Unit IT, University Medical Center of Tübingen, 72076 Tübingen, Germany

**Keywords:** sensor networks, internet of medical things, healthcare monitoring, OCD, telepsychotherapy, WebRTC, security, open standards, performance evaluation

## Abstract

In this manuscript, we describe the soft- and hardware architecture as well as the implementation of a modern Internet of Medical Things (IoMT) system for sensor-assisted telepsychotherapy. It enables telepsychotherapy sessions in which the patient exercises therapy-relevant behaviors in their home environment under the remote supervision of the therapist. Wearable sensor information (electrocardiogram (ECG), movement sensors, and eye tracking) is streamed in real time to the therapist to deliver objective information about specific behavior-triggering situations and the stress level of the patients. We describe the IT infrastructure of the system which uses open standards such as WebRTC and OpenID Connect (OIDC). We also describe the system’s security concept, its container-based deployment, and demonstrate performance analyses. The system is used in the ongoing study SSTeP-KiZ (smart sensor technology in telepsychotherapy for children and adolescents with obsessive-compulsive disorder) and shows sufficient technical performance.

## 1. Introduction

Telepsychotherapy is an innovative method which has been shown to be effective in treating psychological disorders, i.e., patients and therapists hold psychotherapy sessions via online video conferences. A recent meta-analysis indicates that telepsychotherapy is as effective as in-person psychotherapy [1]. Meanwhile, telepsychotherapy is well accepted by therapists [2] and patients [3]. Especially due to the COVID-19 pandemic, telepsychotherapy was even described as the new normal [4] since in-person therapy sessions were not possible during that time.

In addition to the possibility for remote therapeutic conversations, telepsychotherapy also opens up new treatment strategies. For patients with obsessive-compulsive disorder (OCD) [5] which is especially impairing for children and adolescents [6], it is a promising form of therapy because the patient is thereby in his or her home environment, where a large number of the compulsion-triggering situations occur. Studies have shown that behavioral therapies are much more effective and sustainable in the patient’s home environment, where the compulsion-triggering situations occur in daily life, than in artificial clinical therapy situations.

In the SSTeP-KiZ study (smart sensor technology in telepsychotherapy for children and adolescents with OCD), we combine video therapy with wearable sensors to deliver exposure (E) and response prevention (RP) [7] therapy aimed at confronting obsessive-compulsive stimuli at home without performing obsessive-compulsive acts. During E/RP sessions, patients are explicitly exposed to their obsessions while asked to refrain from their usual response and to instead endure the discomfort caused by anxiety or disgust until it subsides on its own.

Within such therapy sessions, wearable sensor technology with eye tracking, electrocardiogram (ECG), and motion sensing allows the therapist to get a better view on the patient’s actions, to gain objective information about the compulsive behavior triggering situations and the patient’s current stress level. This information is very valuable to the therapist for objective therapy evaluation as well as for adjusting the exposure levels during the therapy session. For the realization of such a sensor-assisted telepsychotherapy system, a dedicated hard- and software architecture described in Figure 1 is needed, which realizes the following main tasks.

The synchronous recording of multiple wearable sensors at patient’s home, namely ECG, eye tracking, ego-centric video, and movement sensors.Real-time online streaming and processing of sensor data with an insightful visualization for the therapist.Questionnaire features to measure the therapy progress with gamification features for an intrinsic motivation of filling them out.A secure implementation of data storing and streaming.

**Figure 1 sensors-22-09589-f001:**
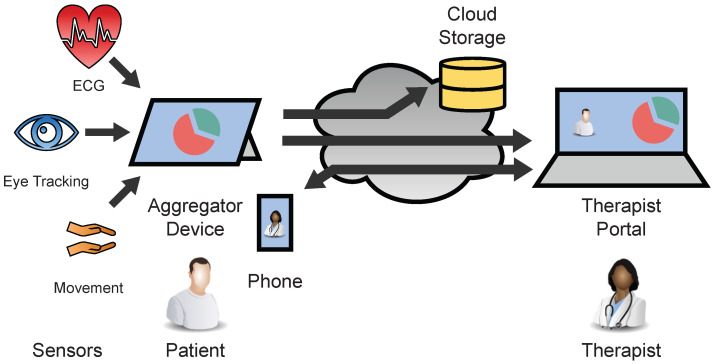
Overview of the SSTeP-KiZ system’s (SKS) infrastructure.

In this manuscript, we describe the soft- and hardware architecture of the SSTeP-KiZ System (SKS) as a modern Internet of Medical Things (IoMT) system. Classic video-telephony is still used in this architecture, but we will not go into its state-of-the-art implementation. The architecture is also checked by the IT security department on medical IT security standards, the data protection department on EU-GDPR compliance, and an ethics commission on ethical aspects. It is actively used in an ongoing feasibility study and our measurements also indicate good performance, but the streaming stability should be improved further.

We start this manuscript with an architecture description, continue with a performance evaluation and some legal aspects about the operation of the infrastructure, and finish with a discussion about the current state. Since this is an interdisciplinary topic, we provide additional information about used technologies in the Appendix to keep this document self-contained and comprehensible even for technical interested but inexperienced readers.

## 2. Architecture

This section describes the architecture of the SSTeP-KiZ System (SKS) with its solutions for

The **transport architecture** for the real-time streaming and the recording of the sensor data including the used sensors and how they transfer their data.The **Questionnaire architecture** where therapists can create questionnaires, send them to the patients and retrieve their responses including gamification elements to motivate the patients.The **administration and security architecture** where administrators can manage the overall system from one place.

We utilize the Vidyo Connect [8] software for the video telephony, which has been used successfully in previous telepsychotherapy studies [7].

### 2.1. Sensor Transport Overview

A general overview of the sensor transport architecture is depicted in Figure 2. It is divided into two pipelines: the Recording Pipeline which stores the sensor data for later access by researchers and therapists, and the Streaming Pipeline which streams the sensor data to the therapist in real time during the telepsychotherapy session.

Each pipeline starts at the sensors of the patient which measure the data and transfer them to the patient’s Aggregator Device. The Aggregator Device processes the data to a desired data format. In case of the Recording Pipeline, the data are cached on the Aggregator Device before they are uploaded to the Storage Server to be accessed by therapists and researchers in the Data Portal. In case of the Streaming Pipeline, the Aggregator Device utilizes the Streaming Server to forward the data to the Therapist Portal in real time to view them to the therapist during the telepsychotherapy session. The Streaming Server thereby consists of three components which will be introduced in Section 2.4.2.

In the following, we describe the Aggregator Device which is the central component of the transport architecture. Then we continue with a description of the sensors which are used in the SSTeP-KiZ System. Finally, we describe the Streaming and Recording Pipeline.

### 2.2. Aggregator

The Aggregator Device is the central component of the transport architecture. It controls the sensors, receives their data, processes the data and forwards them for streaming or recording.

#### 2.2.1. Overview

As depicted in Figure 3, the Aggregator Software is running on the Aggregator Device and is divided into the Aggregator Frontend and the Aggregator Backend. The Aggregator Frontend is a web-based user interface to control the sensors. The Aggregator Backend is a console application which cares about the sensor connectivity and performs the data processing. They are connected to each other via WebRTC, a Real-Time Peer-to-Peer streaming technology for web browsers which is further described in Section A.5.4.

#### 2.2.2. Hardware

As Aggregator Device, we use Microsoft’s Surface Pro 7 with an Intel Core i7-1065G7 processor, 16 GB of LPDDR4X memory, a 256 GB PCIe SSD, Wi-Fi 6 and Bluetooth 5.0 with Windows 11 Pro in version 21H2 (Build 22000.778). The advantage of such a tablet computer is that it can be easily carried in a backpack, has a strong computing power, and can be operated by touch input. To protect the device and the stored health data on it from unauthorized access, we have taken the following measures:We enable BitLocker to encrypt the whole internal storage and verify the OS integrity to prevent booting an eventually manipulated OS.We lock the boot order to prevent booting another OS than the Windows installation.We restricted the UEFI firmware to boot only operating systems signed by Microsoft.We protect access to the UEFI firmware with a strong individual password to prevent unauthorized changes of the boot configuration.To prevent unauthorized logins, the tablets have only two Windows accounts: an Administrator account and a non-administrator patient account.The passwords of both accounts are strong (20 characters consisting of uppercase, lowercase, special letters, and digits) and individual for each device and patient.For a better user experience, patients can log in to their accounts with a 6-digit PIN or via Windows Hello face recognition.We prevent brute-force attacks with the following measures: if the PIN was entered wrong ten times, patients must use their password. After ten failed password tries, the device is locked for 30 min.Patients are forced to use the tablet only at home to add a kind of physical access control layer.Before a tablet is handed over to another patient, we format its disk entirely, clear the TPM’s key storage and reinstall the device with new UEFI and Administrator passwords.

So, the medical health data which are stored on the tablet of a patient are physically protected since an attacker should not be able to enter the patient’s house, and protected by hardened access control mechanisms of the Windows OS. Hence, only our certified Administrators or the patient are able to access the data on the tablet computer.

#### 2.2.3. Backend

The Aggregator Backend Software is a Node.js [9] application which uses the Nest.js [10] framework and runs as a console application on the Aggregator Device. Its main purpose is to receive data of the sensors and process them for streaming or recording. Its architecture is depicted in Figure 4.

The Backend follows a modular “driver”-based infrastructure to connect its sensors. Just like drivers for operating systems, these drivers are software modules which support specific sensor hardware implementations. In our published code, we provide such driver implementations for the sensors described in Section 2.3. These are drivers for one electrocardiograph sensor (ECG driver), one eye tracking sensor (ETK driver) and two different movement sensors (MOV driver). They are depicted on the left of the Aggregator Backend in Figure 4. The drivers receive data from the sensors, process them to an object-oriented representation, and emit them as events to the Aggregation Module. The Aggregation module listens to these events and does the following:If **recording** is on, it forwards the data to the Recording Module. The Recording Module stores the data on the internal storage of the Aggregator Device where the data continue their way through the Recording Pipeline.If there is an active **streaming** connection to a therapist, it forwards the data to the Streaming Module. The Streaming Module awaits incoming streaming connection requests from therapists via a WebSocket connection which is implemented with the Socket.IO [11] library. From there, the data continue their way through the Streaming Pipeline.

#### 2.2.4. Frontend

The Aggregator Frontend is a web application developed with the single-page application framework Angular [12] and acts as a user interface for the Aggregator Backend to control the sensors and view their state as depicted in Figure 5. The Aggregator Backend hosts the Aggregator Frontend locally and the patient runs it in the Chromium-based web browser Microsoft Edge.

To control the devices and update their state, each driver has its own tile in the user interface as depicted in Figure 5. With the buttons on the bottom of the tiles, the patient connects or disconnects the sensors, starts or stops their calibration, and starts or stops the recording. The visibility of these buttons depends on the capabilities of the sensors. The icons on the top right corner of the tile show the recording state (circle), streaming state (waves), calibration state (dot with wave on top), and connection state (link symbol). Again, the visibility of these icons depends on the capabilities of the sensors. The refresh button on the top right of the tile restarts the sensor driver in case of a failure.

To transfer the states from the Backend to the Frontend and to notify the Backend about interactions on the Frontend, the Aggregator Software uses a WebRTC connection. In the Frontend, each tile is connected to a dedicated controller which holds the sensor state. These controllers in the Frontend each have their own Data Channel in the WebRTC connection to the related drivers in the Backend. This communication is depicted in Figure 6.

To improve the precision of the eye tracker’s gaze estimation, the patient must calibrate the eye tracker. This requires the patient to see the eye tracker’s video stream. Therefore, the Aggregator Backend streams the eye tracker’s video stream to the Frontend which displays the video in the eye tracking tile. This is implemented with an additional WebRTC Media Stream between the Backend and the Frontend, which is depicted in Figure 6 as the blue arrow.

### 2.3. Sensors

This section describes the purpose and technical specifications of the used sensors which are shown in Figure 7. The section also describes how the sensors transfer data to the Aggregator.

Table 1 provides an overview of the technical specifications. The used configurations are described in the following related subsections. An overview of the pipelines which describe how the data are transferred from the specific sensors to the Aggregator Device are described in Figure 8.

#### 2.3.1. Movesense HR Sensor—Utilization

We apply the Movesense Sensor HR2 with a chest strap as shown in Figure 7a at the torso of the patient to track the following metrics:The **acceleration** with a 3-axis accelerometer configured with a sampling rate of 52 Hz.The **angular velocity** with a 3-axis gyroscope configured with a sampling rate of 52 Hz.The **magnetic field** with a 3-axis magnetometer configured with a sampling rate of 52 Hz.The heart’s **electrical activity** with a single channel ECG configured with a sampling rate of 250 Hz.

The movement samples will measure the upper body’s movement in relation to the wrist movements which are measured by the APDM movement sensors as described in Section 2.3.3. This helps us to distinguish, e.g., jumps where the whole body moves from pokes where only the hands move. The movement measurement in general is relevant to quantify whether changes in the HRV come only from the patient’s stress level or whether it is also caused by physical activity. This is explained further in Section 2.3.2.

The resulting graphical plot of the measured electrical activity in volt on the y-axis and the time on the x-axis are called electrocardiogram (ECG). According to Kwon et al. [13], the sampling rate should be at least 100 Hz to achieve a moderate data quality, so 250 Hz are a pretty good resolution. A characteristic example of such an ECG voltage series graph is depicted in Figure 9.

The R spike is the one with the largest altitude of an interval. It indicates a heart beat. Our software detects these R peaks from the ECG samples. The time distance between two R waves in milliseconds is called the RR-interval. Our software derives the current heart rate (HR) from the RR interval with Equation (Equation 1) in beats per minute (bpm):(1)$$\mathrm{HR}\;[\mathrm{in}\;\mathrm{bpm}]=\frac{\frac{60000\;\mathrm{ms}}{\;\min}}{\mathrm{RR}\;[\mathrm{in}\;\mathrm{ms}]}.$$

From a sequence of these RR intervals, the software computes the heart rate variability (HRV). The HRV supports the therapist during a therapy session by indicating the patient’s stress level since the HRV highly correlates with the patient’s stress level as follows [14]:High HRV: patient is relaxed.Low HRV: patient is stressed.

This is because under stressed conditions the heart is forced to provide a constant oxygen and blood supply for the body while under relaxed conditions the heart is more flexible.

However, the HRV will also decrease when the body is doing physical activities without being stressed. To prevent such distortions of the stress level’s reliability, we expect to be able to compensate for these distortions by movement measurements as explained in Section 2.3.3.

#### 2.3.2. Movesense HR Sensor—Implementation

All the evaluations described in Section 2.3.1 happen on the Aggregator Device, but the raw ECG voltage values are measured on the sensor and transferred to the Aggregator Device. Since the sensor has no sufficient internal storage, we can only transfer the voltage and movement data via Bluetooth Low Energy (BLE) to the Aggregator Device shortly after their measurement as depicted in Figure 8a. When interference occurs during the BLE transmission, the data will be lost.

BLE provides a standardized interface to exchange information between two devices called the GAP protocol which is described in Section A.1. In GAP, standardized GATT services and characteristics encapsulate the communication interface for, e.g., Heart Rate measurements. Due to missing standardized GATT services and characteristics for movement and ECG data, Movesense has implemented a Bluetooth Classic API to access these data. Since this API transfers the data as JSON-encoded strings, it is not possible to transfer both the ECG and movement data at our preferred sampling rate with Bluetooth’s limited 1 Mbps bandwidth. So we developed a custom firmware for the sensor to provide access to these data at the preferred sampling rate via BLE. Therefore, we defined a custom GATT service with the 16 bit UUID 0x1859 with the following custom GATT characteristics:The **ECG characteristic** with the 16 bit UUID 0x2BDD which contains a set of n=16 measurements, encoded with values concatenated in the following order and as depicted in Figure 10:The relative **timestamp** $t_{n-1}$ as a 32 bit unsigned integer. It indicates how many milliseconds the last sample in the array was measured after startup of the sensor.The **array of voltage samples** with *n* times a 16 bit signed integer. They encode a sequence of ECG voltages in millivolts (mV).The **movement characteristic** with the 16 bit UUID 0x2BE2 which contains a set of n=8 measurements, encoded with values concatenated in the following order and as depicted in Figure 11:The relative **timestamp** $t_{n-1}$ as a 32 bit unsigned integer. It indicates how many milliseconds the last sample was measured after startup of the sensor.The **array of movement samples** with *n* times movement samples containing values in the following order:(a)The **acceleration** to the x, y, and z direction concatenated as 16 bit integer values in centimeter per seconds squared (cm/s${}^{2}$).(b)The **angular velocity** around the x, y, and z axis concatenated as 16 bit integer values in decidegree per second (d${}^{\circ }$/s).(c)The **magnetic field** to the x, y, and z axis concatenated as 16 bit integer values in centigauss (cgauss).

To compute relative timestamps $t_{i}$ for the sample *i* from the relative timestamp $t_{n-1}$, we use the sampling rate *f* (in $Hz=s^{-1}$) and Equation (Equation 2).
(2)$$t_{i}=t_{n}+\frac{i-n+1}{f}$$

As derived from the enumeration of the sensor’s metrics, we insert the ECG sampling frequency $f_{ecg}=250$ and ECG sample length $n_{ecg}=16$, or the Movement sampling frequency $f_{mov}=52$ and the movement sampling length $n_{mov}=8$.

To compute absolute timestamps $t_{i}^{\prime }$ from these relative timestamps $t_{i}$, the offset of the sensor startup time $t_{\delta }$ to the current unix epoch time $t_{unix}$ is estimated with Equation (Equation 3) when the timestamp $t_{n-1}$ is received. Thereby, $t_{\epsilon }$ is the transfer delay caused by the transmission, by buffers, and processing on the sensor and the receiving device. Based on experience values $t_{\epsilon }$ is less than 1 millisecond and can be assumed as $t_{\epsilon }=0$.
(3)$$t_{\delta }=t_{unix}-t_{n-1}-t_{\epsilon }$$

As depicted in Figure 8a, the receiving, parsing, and timestamping is implemented to the Aggregator Frontend in JavaScript. To connect to the sensor via BLE, we used the Web Bluetooth API whose specification is currently in development at the W3C [15].

For Recording purposes, the Aggregator Frontend sends the movement values and the ECG voltages via a dedicated WebRTC Data Channel to the Aggregator Backend. The Aggregator Backend stores the movement samples together with absolute unix timestamps to one CSV file, and the ECG samples together with absolute unix timestamps to another CSV file.

For streaming purposes, the Aggregator Frontend runs a real-time R peak detection. The time offset of two successive R peaks is the RR interval. From a sequence of RR-intervals, the Aggregator Frontend derives the heart rate variability (HRV) with various standardized approaches like SDNN or RMSSD which will not be discussed further in this paper. The Aggregator Frontend transfers these HR and HRV values via a Data Channel to the Aggregator Backend from where they get via the Streaming Pipeline to the Therapist Portal.

#### 2.3.3. APDM Movement Sensors—Utilization

We use one Opal movement sensor by APDM Inc., Portland, OR, USA [16] at each wrist of the patient as shown in Figure 7b to track the following metrics:The **acceleration** with a 3-axis accelerometer at a sampling rate of 128 Hz.The **angular velocity** with a 3-axis gyroscope at a sampling rate of 128 Hz.The **magnetic field** with a 3-axis magnetometer at a sampling rate of 128 Hz.

We use them to track the patient’s physical activity, and to quantify the intensity of the compulsive disorders.

With the APDM movement sensors at the wrists, we can derive the absolute movement energy of the patient’s hand movements. Combined with the movement data from the Movesense sensor at the torso, we can derive the relative movements of the patient’s hands to the patient’s body. This lets us distinguish between, e.g., a jump where the whole body moves which is a stronger physical activity, and a hand raise where only the hands move up and down which is a weaker physical. With the data measured with the SKS, Thierfelder et al. [17] were able to find correlations which improve the reliability of the computed stress level.

The relative hand movements to the torso may also allow a quantification of the intensity of the compulsive disorders, e.g., many patients relieve stress by washing their hands or other compulsive actions. Using the magnitude acceleration vector, one can derive the washing intensity from the acceleration forces. In previous work, these sensors have been used to identify specific disease-related movement characteristics in patients daily life for neurological movement disorders like cerebellar ataxia Ilg et al. [18] and Thierfelder et al. [19].

#### 2.3.4. APDM Movement Sensors—Implementation

For a reliable recording, the APDM Opal sensors store the values on their internal 8 GB flash storage which lasts for about 450 h. To access the recorded raw files, the patient puts the Opal sensors into a USB docking station and connects the docking station via USB to the Aggregator Device as depicted in Figure 8b. As part of the MOV Driver in the Aggregator Backend, we developed a Java application called the APDM Connector Software. This application detects connected APDM Opal sensors, copies new raw data files and converts them to the Hierarchical Data Format v5 (HDF5) which can be used for our evaluations. For the conversion, the application uses the APDM Java SDK since the raw files are a proprietary format of APDM Inc. The converted files are stored on the Aggregator Device’s internal file system from where they continue with the Recording Pipeline.

As soon as we have achieved a reliable stress level estimation, or a reliable compulsive action quantification, we also require real-time access to the movement data to compute the stress level and send it to the therapists. So we prepared the infrastructure to stream the movement data in real time as depicted in Figure 8c. The Opal sensors allow a real-time streaming of their data via a proprietary 2.4 GHz wireless protocol by APDM Inc. The APDM Access Point synchronizes these data and makes them available to the Aggregator Device via USB. Using the APDM Java SDK, the APDM Connector Software accesses these data and streams them as UTF-8 encoded JSON strings to a UDP (see Section A.2.1) endpoint of the MOV Driver in the Aggregator Backend. From there on, the data continue their way through the Streaming Pipeline.

#### 2.3.5. Look! Eye Tracker—Utilization

The eye tracker consists of a field camera placed at the forehead and two eye cameras directed to the patient’s eyes. The field camera allows the therapist to see what the patient sees and the eye cameras detect the patient’s pupils to estimate the patient’s gaze direction.

This is especially relevant in the context of the telepsychotherapy session when the therapist puts the patient into an exposure exercise. Since the therapist can check where the patient is looking, the patient cannot avoid looking at stress-inducing situation during the exercise.

#### 2.3.6. Look! Eye Tracker—Implementation

Since available eye trackers are too large for a children’s head, we 3D-printed an eye tracker as shown in Figure 7c with the following camera properties:Two infrared eye cameras with 30 frames per second and a resolution of 320 by 240 pixels.One field camera with 30 frames per second and a resolution of 640 by 480 pixels.

The eye tracker connects all three cameras via an integrated USB hub with one USB-A cable to the Aggregator Device. For the gaze estimation, we use the Look! eye tracking software as described by Kübler et al. [20]. Therefore, Look! first synchronizes the frames of the three cameras with each other. An example of the three sticked together camera images is depicted in Figure 12. It shows the eye camera images with detected pupils on the right and the field camera with the gaze direction on the left. Then it uses a convolutional neural network (CNN) to extract features like the gaze vector from the pupil positions on the eye camera images. With this gaze vector and parameters from a calibration, the software computes the gaze coordinates which indicate the gaze direction on the field camera.

The Look! software is connected to the ETK driver via WebRTC as depicted in Figure 8d. This connection has one Data Channel to start and stop the calibration and recording, and a Media Stream to receive one video stream which contains the field and the eye camera streams as one. This video stream also renders the gaze estimation as a green circle into the field camera.

For recording purposes, Look! stores this clipped video stream of all three cameras but without the highlighted gaze estimation into one MP4 video file at a resolution of 960 by 480 pixels. Such a sticked-together image is provided in Figure 12. In addition, it stores a TSV file which contains the gaze estimation, timestamps, and other information for debugging. These stored files then continue with the Recording Pipeline. For streaming purposes, the ETK driver simply forwards the video stream with its highlighting from the Look! software to the Streaming Pipeline.

#### 2.3.7. Look! Eye Tracker—Calibration

As described by Kübler et al. [20], the Look! eye tracker computes the gaze estimation with parameters modelling the relation of the detected pupils from the eye cameras and the corresponding two-dimensional position on the field camera. To gain these parameters, it is necessary to calibrate the eye tracker first. Since the accuracy depends on the positioning of the eye tracker on the patient’s face, the patient should calibrate the eye tracker every time he/she puts the eye tracker on.

To calibrate the eye tracker, the patient starts the calibration procedure by clicking on the calibrate button in the Aggregator UI. Then the patient uses a tracker which looks similar to a QR code, focuses with his/her eyes on the tracker, and moves the tracker around the field of view. The eye tracking software detects this tracker pattern on the field camera of the eye tracker and the eye’s pupil position on the eye cameras. Based on the two-dimensional coordinates of the pupils on the eye cameras and the two-dimensional tracker position on the field camera, the Look! software computes the parameters which are necessary to compute the field camera position with the pupil positions on the eye cameras.

The more these gaze position samples are distributed on the field camera that the software has from the calibration procedure, the more accurate are the resulting gaze estimations after the calibration. To give the patient an assessment of the calibrated field of view coverage, the software displays the field camera video during the calibration and darkens the areas of the field video without gaze positioning samples. An example of this calibration video stream is provided in Figure 12.

### 2.4. Streaming Server and Architecture

During the telepsychotherapy session, the therapist initiates a video call with the patient and starts the streaming of the patient’s sensor data. For the video call itself, we use the Vidyo Connect [8] video conference software, whose server is hosted by the medical data center for privacy reasons. To stream the sensor data in real time, we developed the architecture depicted in Figure 13 as described below.

#### 2.4.1. Overview

The sensors stream their data to the patient’s Aggregator Device as described in Section 2.3. The Streaming Module of the Aggregator Backend on the Aggregator Device as depicted in Figure 4 processes these data to a format which is efficient for visualization. Then the Aggregator Device utilizes the Streaming Server to stream sensor data in real time to the therapist’s Therapist Portal. Depending on the network topology between the Aggregator Device and the Therapist Portal, this happens either directly with a peer-to-peer (P2P) connection for better real-time capability, or via the Streaming Server. Once the Therapist Portal receives the sensor data, it visualizes them to be evaluated by the therapist intuitively.

For more data usage transparency, the Aggregator Frontend displays the sensor streaming activities to the patient. Since the patient might wear the Aggregator Device in a backpack, the patient can also watch the sensor streaming activities on the Patient Portal which is optimized for mobile devices like smartphones. To do this in real time, the Aggregator Device streams the information, which therapist accesses which sensor data, in real time to the Patient Portal directly via a P2P connection. This happens like the sensor data streaming but with a dedicated monitoring channel.

To protect sensor data and monitoring information from not permitted users, patients and therapists must authenticate to the Identity Server which is hosted by the medical data center.

#### 2.4.2. Streaming Components

For the connection between the two clients Aggregator Device and Therapist Portal, we utilize WebRTC which utilizes a large stack of real-time protocols described in Section A.5. Hence, we need the following components which were subsumed as the Streaming Server in Figure 13:A **Signalling Server** to which the clients hold a WebSocket connect to exchange session descriptions and ICE candidates.A **STUN Server** which the clients use to gather the ICE candidates and to detect NATs.A **TURN Server** via which the clients can relay their traffic around NATs and firewall restrictions or to hide their IP addresses.

As a Signalling Server, we developed a Node.js [9] application. As a STUN and TURN Server, we used the open source software Coturn [21]. The detailed Streaming Pipeline is depicted in Figure 14.

#### 2.4.3. Security

To restrict the sensor data streaming of patients only to authorized therapists, the Signalling Server forwards the connection requests only for authorized therapists. For the authorization, we utilize the OAuth authorization framework described in Section A.6. To prove authorization, the clients provide their OAuth Access Token when they establish their WebSocket connection to the Signalling Server. This Access Token is issued by the Identity Server and contains information about the patient’s or therapist’s username. The mapping of patients to authorized therapists is held in an internal PostgreSQL database of the Signalling Server. These permissions are modified via the Signalling Server’s REST API. The whole WebSocket connection is protected with TLS 1.3 (see Section A.3.3) whereby the Signalling Server authenticates with a PIK-signed X.509 certificate (see Section A.3.2).

As mandatory in WebRTC, the P2P connection is protected with DTLS 1.3 (see Section A.3.4). The mutual authentication of the clients happens with self-generated X.509 certificates which are exchanged in the session descriptions via the Signalling Server. To protect also meta data at the Coturn server for STUN or TURN, we configured the server to force DTLS traffic and to authenticate with a PKI-signex X.509 certificate. Since DTLS 1.3 is not yet supported by Coturn, we force DTLS 1.2 which is specified in RFC 6347 [22]. We further restricted Coturn to use only cipher suites which are considered as secure by the National Security Agency (NSA) of the United States government in RFC 9151 [23] and by the Federal Office for Security in Information Technology (BSI) of the German government [24].

#### 2.4.4. Sensor Data Channels

The communication inside the WebRTC connection is structured with dedicated channels per sensor as depicted in Figure 15.

By exchanging session descriptions via the Signalling Channel, these channels are configured as follows:The **eye tracking** video is streamed with RTP via UDP with a Media Stream. The Therapist Portal directly displays this video.The **ECG** evaluations are streamed JSON-encoded via SCTP with a Data Channel. The Data Channel has a maximum packet lifetime of 1 sec and does not care about ordered communication since the ECG values are timestamped so that the correct order can be reconstructed. The Therapist Portal visualizes the HR and HRV values as diagrams.The **movement** values are streamed JSON-encoded via SCTP with a Data Channel. The Data Channel has a maximum packet lifetime of 1 sec and does not care about ordered communication since the movement values are timestamped so that the correct order can be reconstructed. The Therapist Portal does not visualize these values yet since this would not be helpful for the therapists but we expect that the machine learning evaluations will help us to detect and display anomalies to the therapist in future.

#### 2.4.5. Therapist Portal

When the Therapist Portal receives sensor data, it visualizes them as in the screenshot of Figure 16.

In the background, it renders the eye tracking video stream. This helps the therapists to see what the patient sees and where the patient looks at. On the top right, the Therapist Portal plots the actual and the mean heart rate to a live line chart. Below, it does the same thing with the HRV. The pie chart on the bottom right shows the proportions of the high (HF), low (Low) and very low (VLF) frequencies in the heart rate. The therapists use these charts to see the stress level of the patient. Therefore, the therapists start a baseline measurement with the buttons below the charts. This is done at the beginning of every therapy session. If the HRV decreases, which also results in a higher VLF proportion, the patient is more stressed than in the baseline measurement. If the HRV increases, the patient is more relaxed. The buttons and the text field on the bottom allow the therapist to label the current actions. These labels are important for the offline analysis of stress levels during the exposure exercises.

The application itself is a web application which we developed with the Angular [12] Framework. We host its HTML, CSS and JavaScript file as static files with an Nginx [25] HTTP server via HTTP/2 (see Section A.4).

### 2.5. Recording Architecture

As described in Section 2.3, the sensors store their data on the Aggregator Device’s internal storage. This section describes their further way via the Storage Server to the Data Portal.

#### 2.5.1. Overview

Therapists and researchers use the Data Portal to download the recorded data from the Storage Server to recap therapy sessions and for future research like the ML-based evaluation of the movement data. Before patients can upload sensor data and therapists can access them, they must authenticate to the Identity Server. Figure 17 depicts this entire Recording Architecture.

#### 2.5.2. Storage Server

The Storage Server is a WebDAV server with a large storage capacity. We use the open source Nextcloud application [26] for that. The Aggregator Device uploads its data with the HTTP-based WebDAV protocol (see Section A.4) to the server. Therefore, we utilize the Nextcloud Sync Client which watches a local directory on the Aggregator Device, uploads new files to the Nextcloud Server, and removes uploaded files from the internal storage to save storage capacity on the Aggregator Device. To provide therapists access to the uploaded sensor data files, the patient shares them with the integrated folder sharing feature of Nextcloud. The therapists can then browse these files by using Nextcloud’s Web-based user interface whose screenshot is provided in Figure 18. We call this user interface the Data Portal.

The Storage Server is equipped with ten 10 TB disks in two RAID-6 bonds. So it has about two times 30 TB of usable storage whereby up to two disks per bond may fail without losing data and without impact on availability. For GDPR compliance and regulatory reasons, the server hardware is located in an ISO-27001 certified data center and the Nextcloud Enterprise software is managed by an ISO-27001 certified service provider.

#### 2.5.3. Security

The data transfers to and from the Storage Server are protected with TLS 1.3 (see Section A.3.3). The Storage Server thereby authenticates with a PKI-signed X.509 certificate (see Section A.3.2). Patients and therapists log in to the Storage Server via OpenID Connect (see Section A.6.4). Therefore, the users sign in to their account on the Identity Server. The Identity Server issues an ID Token which is used to authenticate Nextcloud’s internal OAuth Authorization Server. This Authorization Server issues an OAuth Access Token which the Nextcloud Sync Client on the Aggregator Device and the Data Portal use to up- or download the sensor data files.

To protect the stored data on the Nextcloud server from unauthorized access, we run it on a hardened Linux-based operating system. SSH access is only possible via key-based authentication from a restricted IP address space of the university. Nextcloud is configured to provide access to its stored data only to authenticated users. To further restrict access to a patient’s data only to the patient’s therapist, we use Nextcloud’s built-in access control mechanisms via its share feature. To authorize a therapist to access its patient’s data, the patient must explicitly share its data folder with the therapist.

### 2.6. Questionnaire Architecture

The therapists regularly ask questionnaires with their patients to evaluate the psychotherapy progress. Patients should thereby fill out the questionnaires in a timely manner since they contain questions about their feelings over the current day. To motivate the patients to do this properly, they get rewarded for proper submissions with gamification elements in the Patient Portal. This is an effective tool to increase the response rate [27].

#### 2.6.1. Overview

A typical workflow of the questionnaire feature looks like this:Therapists **create questionnaires** in the Therapist Portal.The Therapist Portal uploads these questionnaires to the IMeRa Server which stores structured study data of the medical center.Patients **answer questionnaires** in the Patient Portal.The Patient Portal therefore downloads the questions from the IMeRa Server and uploads the answers.Therapists **evaluate answers** in the Therapist Portal.The Therapist Portal therefore downloads the answers from the IMeRa Server

The communication of this workflow is depicted in Figure 19.

#### 2.6.2. IMeRa Server

The Integrated Mobile Health Research Server (IMeRa) [28] stores the questions and answers of the therapist’s questionnaires. Therefore, the Patient and Therapist Portal communicate via a REST API with the Server which is written in Java with the Spring framework [29] and stores the data in a Microsoft SQL database. The university medical centre’s IT department hosts this server in their data center. To protect the data, the server uses TLS 1.3 (see Section A.3.3) and authenticates with a PKI-signed X.509 certificate (see Section A.3.2).

#### 2.6.3. Patient Portal

The Patient Portal is the web application for patients to fill out questionnaires and to report their well-being. It also implements the UI for the gamification feature which is described further in Section 2.6.4. Since the portal will be used by children, the user interface and illustrations were designed specifically for children for a better end user oriented user experience.

The web application is developed with the Angular framework and is hosted via HTTP/2 (see Section A.4) with an Nginx server. It accesses the gamification progress and handles in-game purchases via the Gamification Server’s REST API. It also downloads questionnaires and uploads answers and well-being feedback from and to the IMeRa REST API.

#### 2.6.4. Gamification

It is often very stressful for patients to answer daily and weekly questionnaires about their emotional state and symptom severity. For this reason, we developed gamification features to serve as an incentive for the children to complete the questionnaires. The core idea is that patients can design their own avatar in the Patient Portal and receive coins for each questionnaire they answered. These coins can then be used to design the avatar (clothes, accessories) and for traveling around the game. The more questionnaires the patient answers, the more areas of the game (continents) are unlocked for the patient. Figure 20 shows an example of the graphic style in a screenshot.

To persist the game state of each patient across devices, we developed a Gamification Server. This Gamification Server stores the current progress of the level, bought items, and the money balance in a PostgreSQL database. It also provides a REST API to access the state, buy items, and to manage the patient profiles. It is implemented with Node.js and the Nest.js framework.

### 2.7. Administration and Security

This section describes the overall administration and security architecture. It describes the Identity Server, introduces the Administrator Portal, and explains how the infrastructure is maintained in a containerized environment (see Section A.7.2).

#### 2.7.1. Identity Server

The Identity Server is an OAuth Authorization Server and an OIDC OpenID Provider at once (see Section A.6). As implementation, we use the open source software Keycloak [30] with a Microsoft SQL Database as user storage. Keycloak provides a web-based user interface for administrators to

**Configure** specific scopes for the Signalling Server, IMeRa Server, and Gamification Server, which act as **OAuth Resource Servers**,**Register** the Therapist Portal, Patient Portal, Aggregator Software, and the Administrator Portal as **OAuth Clients**, and**Manage** the patient’s and therapist’s **user accounts**.

To improve the security, users authenticate with their usernames and a password-less TPM authentication using W3C’s WebAuthN standard [31]. This is a secure and user-friendly authentication method which is safe against phishing attacks where attackers try to phish the user’s credentials via faked authentication pages. Therefore, e.g., the patient’s Aggregator Device is pre-registered for the patient’s user account. When the patient wants to log in to the Aggregator Software, he can use his Windows PIN or face instead of entering his password.

Since the Authorization Server (AS) is the root of trust of all the Resource Servers when it comes to access control, the AS is the most critical infrastructure component. Therefore, the AS is hosted in the university medical center’s data center by the IT department and also authenticates with a PKI-signed X.509 certificate (see Section A.3.2).

#### 2.7.2. Administration Portal

The following list provides a summary of what the administrator must set up to add a user to the study, and to delete when the user leaves the study:The user account on the Authorization Server.The user’s profile on the Gamification Server.The user’s profile on the IMeRa platform.The user’s streaming permissions on the Signalling Server.

Since it is not user-friendly to manage this via REST APIs or via different UIs, we developed the Administration Portal. This is an Angular web application which utilizes the REST APIs of all the related servers to provide a single and clear management interface and is hosted via HTTP/2 (see Section A.4) with an Nginx server. A screenshot is provided in Figure 21.

#### 2.7.3. Server Administration

Table 2 provides an overview of all involved servers and the services running on them. The IMeRa and Keycloak server are both hosted by the IT department of the university medical center for legal reasons. The Nextcloud server is hosted by an external service provider for legal reasons, too. The remaining server hosts many different services.

To simplify the management of these services, each service is containerized in a separate Docker [32] container. This improves the security of the environment due to the service isolation and also improves the maintainability. To deploy them all at once, the entire composition is orchestrated with Docker Compose [33] where environment-specific variables are configured in a simple .env file as environment variables. This also allows us to start and stop individual containers or all at once with a single command. A bash script installs the required components and pulls the git repository from an external git server. The image-based container deployment, its configuration file based orchestration, and the automated script-based deployment implements the Infrastructure-as-Code principle. This lets us (re-)deploy the entire infrastructure fully automated on any other server.

#### 2.7.4. Reverse Proxy

The Signalling and Gamification Server both provide an HTTP-based REST interface (see Section A.4). Additionally, the Admin, Therapist, and Patient Portal are accessed via HTTP. To make them secure, TLS (see Section A.3.3) is used, and several HTTP security headers are set. It is an elaborate process to configure this for each HTTP service and also increases the number of attack surfaces from the Internet. A centralized Reverse Proxy which serves as a single entrypoint from the Internet, as depicted in the infrastructure graph in Figure 22, is a best common practice to solve this. Due to its good integration to Docker Compose, this is implemented with the Traefik Proxy [34]. Traefik thereby acts like an application-layer gateway which forwards HTTP requests from the Internet to the internal Docker Compose network. It thereby handles HTTPS traffic from the Internet automatically and manages the X.509 certificates, issued by Let’s Encrypt, and proxies the HTTPS requests via internal HTTP connections. It also adds HTTP security headers and redirects HTTP to HTTPS requests to enforce a TLS 1.3-protected communication. This is also configured in the Docker Compose orchestration file.

#### 2.7.5. Client Administration

In addition to the Backend Services, also the patient’s Aggregator Device must be administrated. This includes the installation of the Aggregator Software, the APDM Connector software, the Look! Eye Tracking software, the Nextcloud Sync Client, and regular updates. A PowerShell script which installs the required programs and updates, and also configures most of the Windows system, solves this automatically. Afterwards, only a few configurations must be set manually, such as the BitLocker drive encryption, the user account creation, and the Nextcloud Sync Client configuration.

## 3. Evaluations

In this section, we evaluate the storage requirements by sensors, provide a performance evaluation of the Aggregator Device to estimate the minimum hardware requirements, and a network utilization evaluation to estimate the minimum network requirements. We did the performance evaluation with the infrastructure deployment that we use for the SSTeP-KiZ study whose components are described in Section 2.7.3. Since this evaluation targets the performance of the technical infrastructure, no patient data are involved. The measurements were carried out under laboratory conditions by the authors but the feedback of our therapists showed that they are comparable with real-world scenarios from the study.

### 3.1. Storage Requirements

To estimate the storage requirements over time of each sensor, we took the recorded file sizes and normalized them to Megabyte per hour (MBph). The results are shown in Table 3.

According to the percentages, one can clearly see that the video recording of the eye tracking consumes by far most of the storage (97.3%). We repeated this evaluation and found out that all these values were the same, except the storage requirements of the video recording. The reason for that is that the file size of the video recording depends on the video content and the MPEG-4 codec’s ability to compress the video.

### 3.2. Aggregator System Performance

To estimate the minimum system requirements of the Aggregator Software, we measured several performance indicators with Windows 11’s built-in Performance Monitor at a sampling rate of 12 samples per minute while recording the files from Section 3.1. The measurement results are compiled in Figure 23 and discussed in the following.

#### 3.2.1. Memory Utilization

Figure 23a shows the memory utilization of the Aggregator Device during the recording and streaming session. The 64-bit Windows 11 system already utilizes about 4.0 GB of memory from the beginning. Another 1.6 GB are required after startup of the Aggregator Software which results in the first peak of 5.6 GB. The average memory utilization is 5.3 GB which is utilized constantly over the runtime of the Aggregator Software. Hence, we recommend a Windows 11 ×64 operating system with at least 6.0 GB of memory as Aggregator Device.

#### 3.2.2. CPU Utilization

Figure 23b shows the CPU utilization of the Aggregator Device during the recording and streaming session. It shows that the Intel Core i7-1065G7’s base clock rate of 1.3 GHz (100%) is overclocked with Intel’s Turbo Boost technology during the entire therapy session. Starting with a peak of 3.1 GHz (239.7%) for a fast startup, the processor holds an average clock rate of 2.3 GHz (175%) over its 4 cores with 2 threads each. Hence we recommend running the Aggregator Software on a device with a CPU of at least 2.4 GHz and 8 threads.

Right after connecting the eye tracker, the device’s active cooling fan becomes noticeable. The evaluation results in Table 4 show that the Look! Eye Tracking software takes 56.83% of the total CPU power while the Aggregator Backend and the Aggregator Frontend consume together only 39.37%. Since we realized in previous measurements that the Nextcloud Sync Client consumed approximately 10% of the CPU time for detecting file changes in the subscribed directory, we turned it off during the recording. The majority of the other background tasks come from the Performance Monitor which recorded the measurement, so this is negligible. Additionally, the Windows Defender consumes 1.02% but for security reasons we keep it on.

#### 3.2.3. Energy Consumption

The recording and streaming started with a full battery of 43.0 Wh, which indicates that the one year old battery has 92% of its factory-new 46.5 Wh capacity remaining. After the 1:33:30 h recording and streaming, the battery was at 2.7 Wh (6%). This means that the Surface Pro 7 with an Intel Core i7 processor consumes $\frac{(43.0-2.7)\;\mathrm{Wh}}{1.558\;h}=25.9\;W$ to run the Aggregator Software with all sensors in recording and streaming state. Compared to the Surface Pro 7’s promised 10.5 hours of battery life, 1.558 h are not enough, but sufficient for a typically one hour telepsychotherapy session.

#### 3.2.4. I/O Disk Utilization

Figure 23c visualizes the read and write speed of the Aggregator Device’s internal SSD. Except for a few read operations when starting the applications and probably some background tasks, the software does not read any data from the disk. In contrast, the write speed is constantly at 0.8 MBps, which is mainly due to the video recording.

### 3.3. Bandwidth Consumption

The evaluation of bandwidth consumption is twofold. On the one hand, we measure the consumed bandwidth without any traffic shaping to estimate the recommended bandwidth requirements. On the other hand, we limit the available bandwidth to estimate the minimum bandwidth required for sufficiently high video quality.

#### 3.3.1. Bandwidth Consumption without Limitations

To estimate the recommended bandwidth requirements, we measured the bandwidth consumption during the recording and streaming in the previous sections. Thereby, we did not apply any traffic shaping. Figure 23d shows the utilization of the Wi-Fi network interface of the Aggregator Device. It shows that the downstream rate is negligible and probably comes from the WebSocket connection and background tasks. The mean upstream bitrate is 2.2 Megabits per second (Mbps). Since a short peak of 3.4 Mbps happens only at the beginning and a value of 2.5 Mbps was reached only six times, we declare this as the recommended available upload bandwidth on the patient side, and as the recommended download bandwidth on the therapist side.

#### 3.3.2. Bandwidth Consumption with Limitations

To estimate the minimum required bandwidth, we used the Software NetLimiter 5 [35] to limit the upload bandwidth step-wise from 2000 via 1000, 500, 250, 125, and 64 kilobits per seconds (kbps). Since the download bandwidth of the streaming is negligible, we did not limit the download bandwidth. During the step-wise restriction of the upload bandwidth, we observed the data stream in the Therapist Portal. We also recorded the network utilization with Windows’ built-in Performance Monitor at 12 samples per minute to verify the effectiveness of the upload limit. In addition, we measured WebRTC’s built-in video streaming statistics to collect quantitative values about the received video’s quality. The result is depicted in Figure 24.

While the video resolution did not change with the bandwidth reduction, every time the bandwidth was reduced, the frame rate dropped before WebRTC adapted to the new available bandwidth and recovered the connection. While a limitation from unlimited down to 500 kbps just led to short instabilities of the frame rate, a further reduction to 250 kbps reduced the frame rate from 30 fps to about 15 fps which is still tolerable. The next reduction step to 125 kbps reduced the frame rate even further to about 5 fps which is noticeably lagging. At 64 kbps, the frame rate went below 1 fps which is unusable. After stopping the traffic shaping, WebRTC started to exponentially increase its quality and the related bandwidth. From our observations, we recommend a minimum upload bandwidth of 500 kbps.

## 4. Summary

In this section, we describe how medical professionals use the system by an example of a typical telepsychotherapy session and what measures we took to ensure that the software can be legally used with real patients.

### 4.1. Application

The system was primarily developed for psychotherapists to treat children with compulsive disorders and it is currently used for research purposes to improve future treatments. This section describes the workflow of such a telepsychotherapy session and the questionnaires from a summarized technical and medical perspective.

Before a regular telepsychotherapy session starts, the patient charges the tablet’s battery to ensure that the battery lifetime is enough for a typical one-hour telepsychotherapy session. The therapist and the patient start the session by entering the Vidyo Connect video conference room. The patient then puts the eye tracker on his/her face, the APDM movement sensors on his/her wrists, and the Movesense ECG on his/her breast, connects the sensors to the Aggregator Device via the various interfaces as depicted on the left of Figure 25 (USB for the eye tracker, BLE for the ECG, and proprietary 2.4 GHz wireless via the APDM Access Point and USB to the Aggregator Device). When the patient starts the Aggregator Software, its Backend connects to the Signalling Server with a bidirectional WebSocket connection as depicted in Figure 25 with the purple double arrow. Then the therapist opens the Therapist Portal which also establishes a WebSocket connection to the Signalling Server so that the therapist can call the patient. The Signalling Server forwards the call request to the Aggregator Software which then starts the WebRTC P2P call with the Therapist Portal. The Aggregator Software therefore utilizes the STUN server and may tunnel the P2P connection via the TURN server which is depicted in Figure 25 with the black arrows. Via this encrypted connection, the therapist receives the sensor data which are visualized in its Therapist Portal.

The transmission of the eye tracking data in real time allows the therapist to capture what the patient sees from the first-person perspective with the gaze direction highlighted. The therapist uses this information to check whether the patient is actively facing his/her fears or avoiding them, for example by looking at the floor instead of the triggering stimulus and not actively perceiving the object with his/her eyes. By evaluating the heart rate (HR), the therapist sees a trend in how anxious the patient is compared to the baseline at rest. An increased heart rate often correlates with a more regular heart beat which results in a decreased heart rate variability (HRV) and is associated with an increasing stress level when the patient has to face his/her fears during the exercises in the therapy session. This allows the therapist to assess how the patient is adapting to the stress level that the therapist is increasing through the chosen exercises.

The therapist can also identify when the stress level becomes too high so that the exercises need to be modified or discontinued. This assessment is difficult even when the patient and therapist are in the same room, and nearly impossible when the therapist can only observe the patient via webcam. With the added sensors, the ability to assess the patient’s tension improves. However, therapists need to be trained to interpret the data correctly. During the session, the therapist sets various tags to transmit the timestamps of the ongoing exposures to the IMeRa server. This allows precise interpretation of the data depending on the session content when researchers evaluate the therapy session afterwards.

After the telepsychotherapy session, the Nextcloud Sync client uploads the recorded medical data from the session to the Storage Server via WebDAV so that therapists and researchers can access and combine them with the tags from the IMeRa server. These labels and ECG data help researchers and therapists better understand how constraints affect tension levels. We are also using the recorded motion data to combine them with the ECG data to derive an indication of the patient’s stress level [17]. Once this is reliable, we will also incorporate it into real-time streaming.

The therapist also creates questionnaires and uploads them to the IMeRa server where patients access and answer them via HTTP on the patient portal. The patients then upload the answers to the IMeRa server where the therapist can download them from for an overview of the therapy progress. To motivate patients to answer these questionnaires, a reinforcement game was implemented into the app. In this game, patients earn coins with each completed questionnaire. With the earned coins, patients can buy clothes for their own avatar and unlock new levels. The patient portal saves this game progress via HTTP on the Gamification Server.

### 4.2. Legal Aspects

As the SKS is used in an ongoing telepsychotherapy study with real patients, it must comply with many legal requirements.

The SKS must be compliant with the general **data protection** regulation (GDPR) of the European Union. Therefore, we pseudonymized all sensor data since they are personal medical data and must be protected. The SKS also passed a check of the data protection department of the university medical center before it was allowed to be used for patients.

The SKS also passed an **IT security** check by the university medical center’s department. This resulted in the requirement that ISO-27001 certified data centers and service providers are needed to host the Storage Server. We also had to configure the WebAuthN authentication to protect the user accounts while keeping the login process user-friendly.

The SKS passed an **ethical** check which effected some changes. Originally, a leader board for the gamification feature was planned so that patients could compare their scores from the questionnaires to those of other patients. However, from an ethical perspective it is not recommended to make the curing of children with compulsive disorders to a challenge and put them under additional pressure. So we discarded that idea to pass the check of the ethics commission.

From a **physical** perspective, we had to keep track on the safety of the patients. Since we 3D-printed our eye trackers and use custom camera systems for them, we had to make sure that especially the infrared LEDs do not harm the patient’s eyes with their radiation. Therefore, we had to use modelling clay to prevent that the LEDs get too close to the patient’s eyes.

## 5. Conclusions

The aim of this software project was to develop the architecture of the SSTeP-KiZ System (SKS), an Internet of Medical Things (IoMT) system for telepsychotherapy. The system supports therapists during the behavioral exposure sessions with sensor information from the patient in real time. For this purpose, eye tracking, ECG, and movement data were used. The eye tracking provides insights into the patient’s gaze direction during the session which gives the therapist better knowledge about the patient’s real behaviour than a normal video call can do. The ECG, namely the derived HRV data, provides insights into the patient’s stress level which lets the therapist quantify the strength of the compulsive exposure. The movement measurement provides insights into the patient’s physical activity which improves the accuracy of the stress level values and lets the therapist quantify the strength of the patient’s compulsive actions.

For that purpose, we designed the SKS as a modular system which is capable of connecting wearable sensors via Bluetooth Low Energy, USB, and via a network connection to the Aggregator Device. To stream the sensor data securely from that Aggregator Device to the Therapist Portal, we utilized the open and modern web standard WebRTC. We forced the use of high security standards, and demonstrated that the transmission should also work well even from homes with slow Internet connections. Nevertheless, we encountered disconnects when the WiFi connection breaks or is unstable. This often resulted in complete disconnects so we had to restart the Aggregator Device which interrupted the therapy session flow.

In addition, SKS is capable of recording the gathered sensor data in full quality and upload them to a centralized server using the open web standard WebDAV. From there, the data can be accessed by HTTP-based web interfaces. For a better progress indication, we also implemented a questionnaire feature. Gamification elements were added to motivate patients to fill out the questionnaires. The SKS leverages the modern authorization framework OAuth 2 to provide centralized role-based access control mechanisms and Single Sign-On (SSO) capabilities. To facilitate setup and maintenance, SKS is deployed in a containerized environment. Since the SKS is utilized in practice for the treatment of patients, it has to comply with various requirements with regard to IT security, data protection, and ethics which are all acknowledged by the responsible department.

In the ongoing feasibility study, we are verifying whether the system delivers the expected treatment improvements. We also collect the sensor data to be able to verify the concept of the stress level measurement and to improve the reliability of the stress level estimation. In future studies, the aim is to increase the self-empowerment in real-life interventions and exercises let patients do their therapy exercises more independently with therapeutic feedback in separate feedback sessions. There are also plans to use the system for adults, but this will require an adaptation of the user interfaces currently designed for children. In general, the effectiveness of such a self-empowered therapy strategy should be investigated and improved further since it has the potential to improve the overall treatment of psychological diseases with less therapeutic effort.

We made the source code for the Aggregator Software, the Backends, and the Frontends of SKS publicly available on GitHub [36] as well as the custom Movesense ECG device firmware [37].

## Figures and Tables

**Figure 2 sensors-22-09589-f002:**
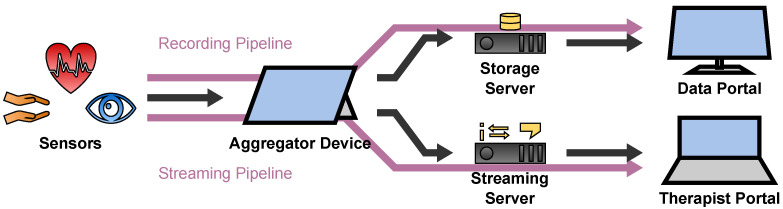
Overview of the sensor transport architecture.

**Figure 3 sensors-22-09589-f003:**
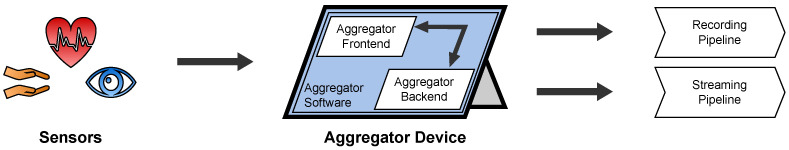
Overview of the Aggregator Device.

**Figure 4 sensors-22-09589-f004:**
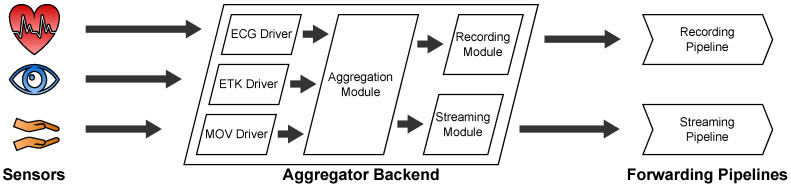
Architecture of the Aggregator Backend.

**Figure 5 sensors-22-09589-f005:**
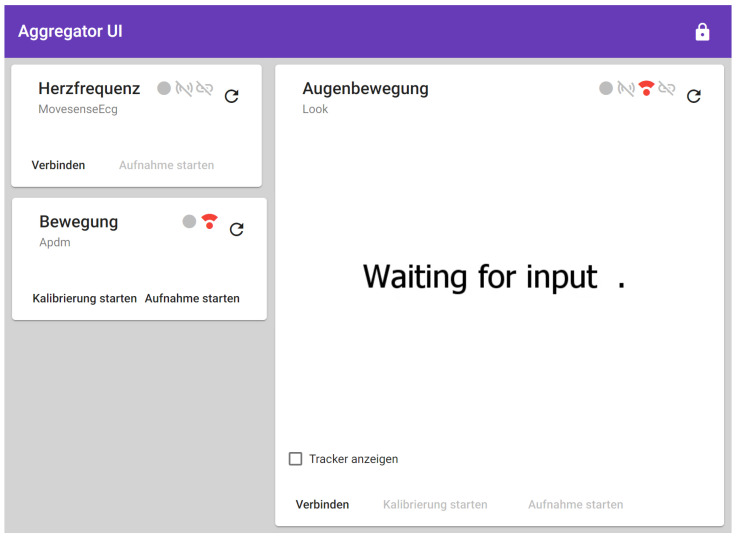
Aggregator Frontend with control tiles for each sensor driver.

**Figure 6 sensors-22-09589-f006:**
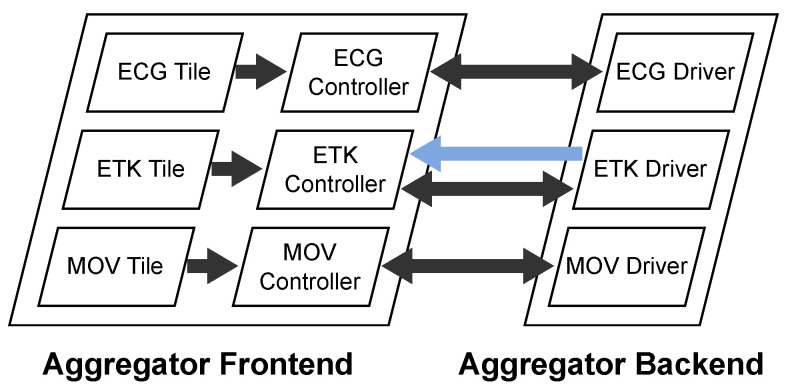
Frontend-to-Backend communication within the Aggregator Device.

**Figure 7 sensors-22-09589-f007:**
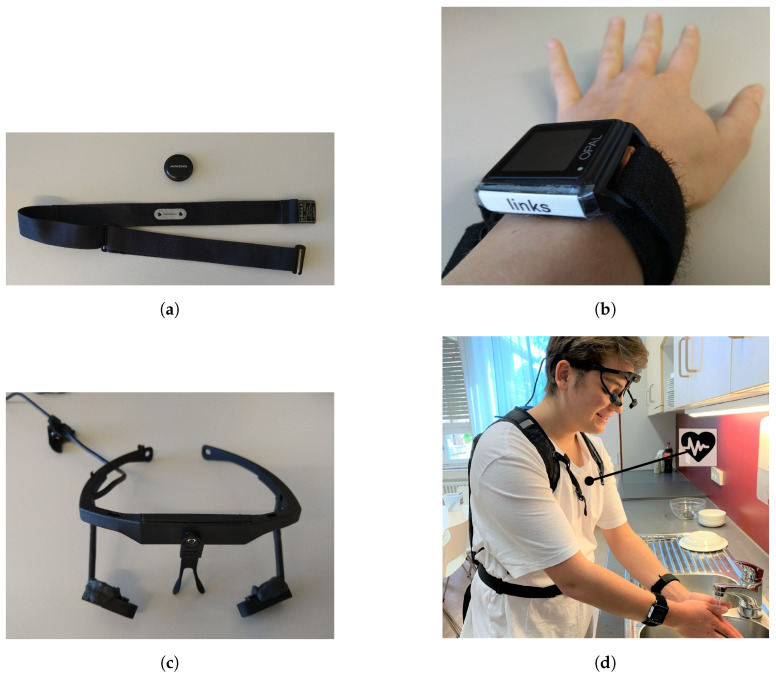
Illustrations of all sensors applied in SKS. (**a**) Movesense ECG and movement sensor with breast belt. (**b**) One of two APDM Opal Movement Sensors on the left hand. (**c**) The 3D printed Look! eye tracker. (**d**) All sensors at a patient.

**Figure 8 sensors-22-09589-f008:**
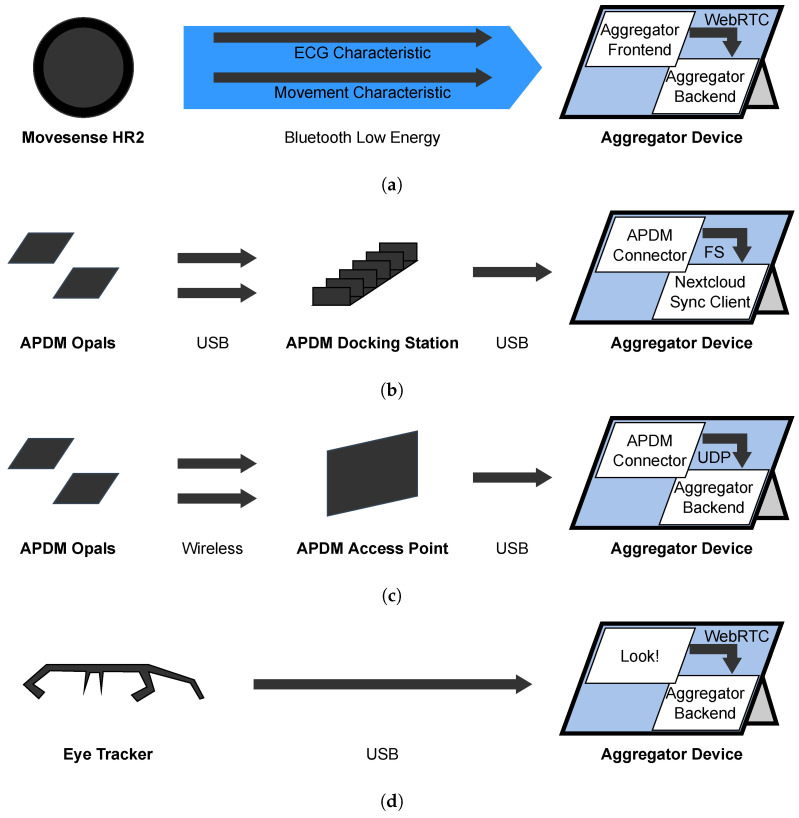
Overview of sensor Streaming and Recording Pipelines. (**a**) Movesense movement and ECG Streaming and Recording Pipeline. (**b**) APDM sensor Recording Pipeline. (**c**) APDM sensor Streaming Pipeline. (**d**) Eye Tracking Streaming and Recording Pipeline.

**Figure 9 sensors-22-09589-f009:**
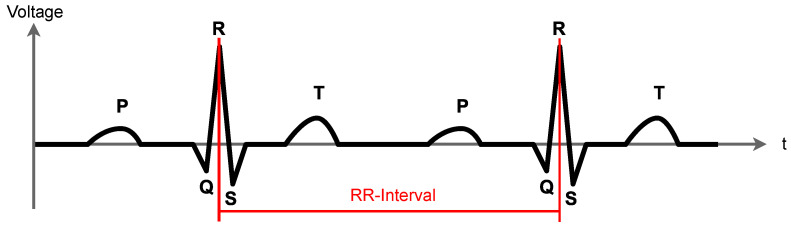
Characteristic ECG voltage curve with an RR interval in red.

**Figure 10 sensors-22-09589-f010:**
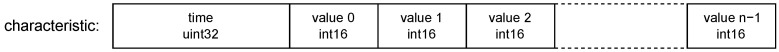
Encoding of ECG characteristics.

**Figure 11 sensors-22-09589-f011:**
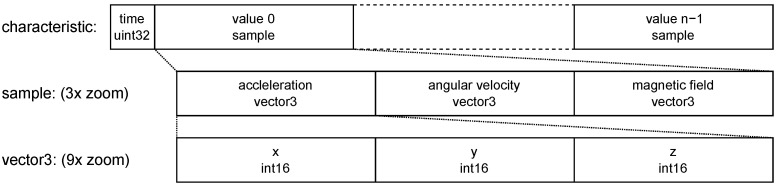
Encoding of movement characteristics.

**Figure 12 sensors-22-09589-f012:**
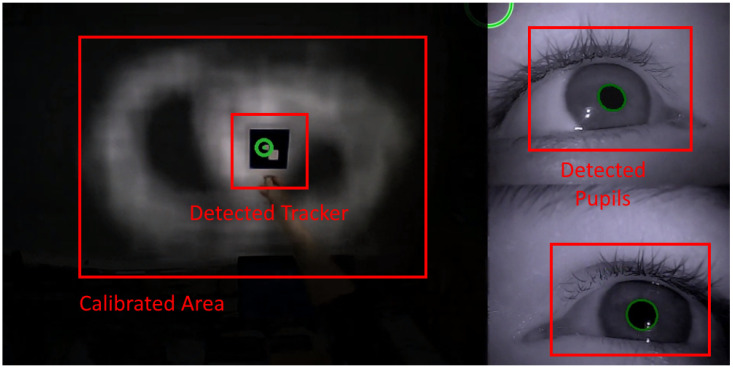
Screenshot of eye tracking calibration. Green labels are displayed by the eye tracking software to the therapist, rectangular red labels are inserted to mark concepts described in the text.

**Figure 13 sensors-22-09589-f013:**
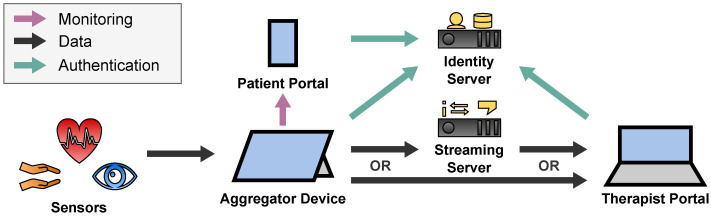
Streaming Architecture of the SKS.

**Figure 14 sensors-22-09589-f014:**
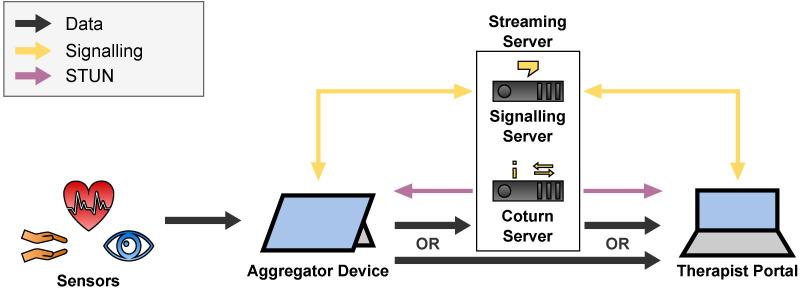
Detailed streaming pipeline including components of Streaming Server, consisting of the Signalling and STUN/TURN Server.

**Figure 15 sensors-22-09589-f015:**
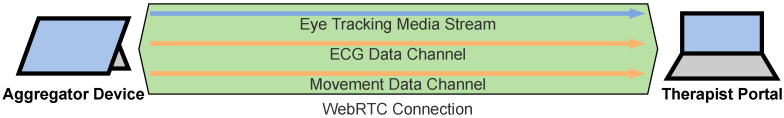
Sensor data are carried from the Aggregator Device to the Therapist Portal within Media Streams and Data Channels of a P2P WebRTC connection.

**Figure 16 sensors-22-09589-f016:**
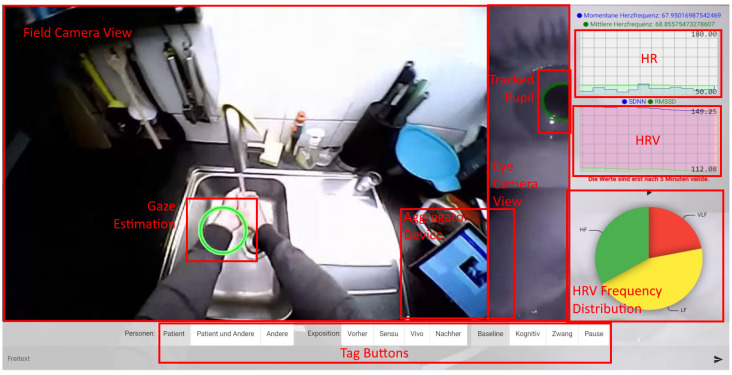
Therapist Portal with sensor data streaming in action. Green labels are displayed to therapist, rectangular red labels are inserted to mark concepts described in the text.

**Figure 17 sensors-22-09589-f017:**
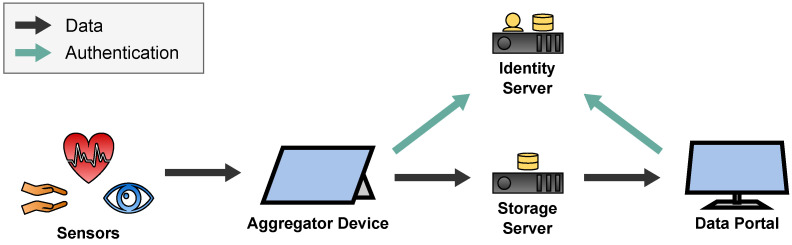
Recording Architecture of the SKS.

**Figure 18 sensors-22-09589-f018:**
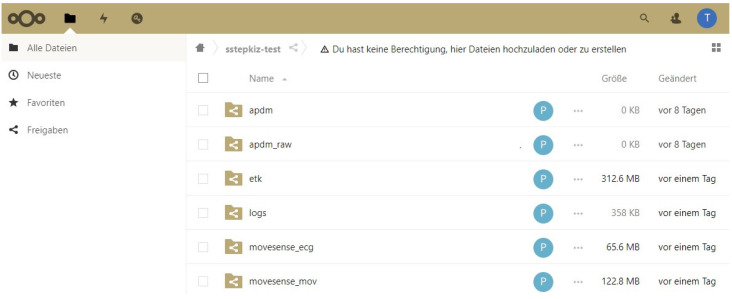
Data Portal: the Nextcloud web interface with sensor-specific folders of a patient.

**Figure 19 sensors-22-09589-f019:**
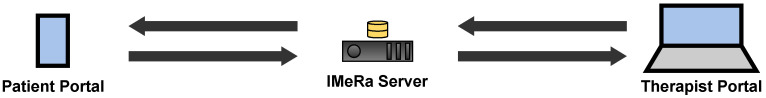
Architecture of the questionnaire feature.

**Figure 20 sensors-22-09589-f020:**
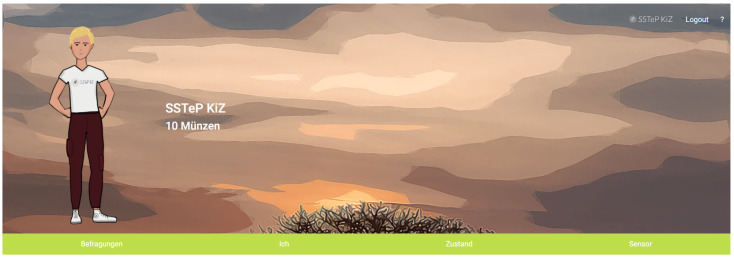
Patient Portal: screenshot of avatar with its clothes and accessories in South America.

**Figure 21 sensors-22-09589-f021:**
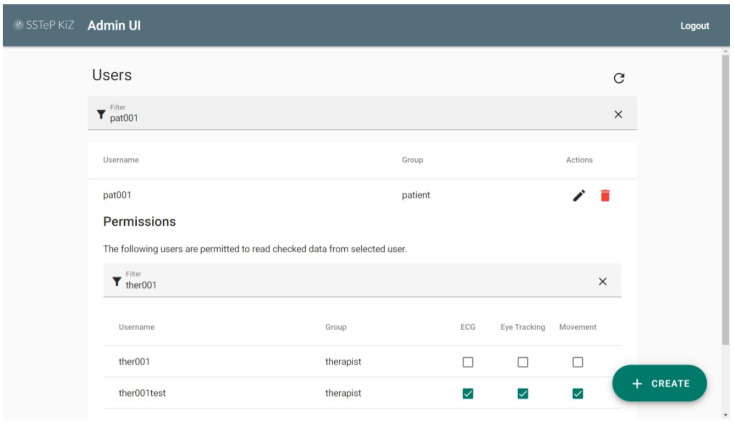
Screenshot of the Administration Portal.

**Figure 22 sensors-22-09589-f022:**
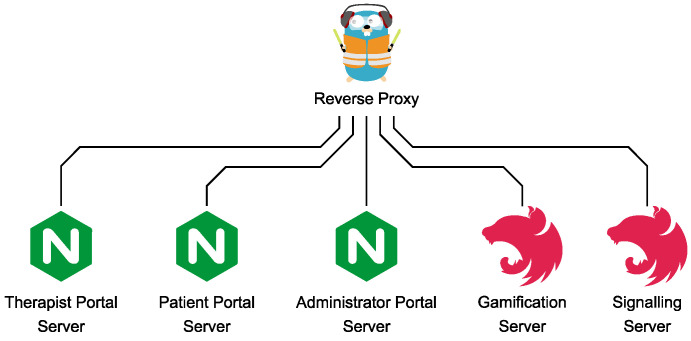
Reverse Proxy with proxied services.

**Figure 23 sensors-22-09589-f023:**
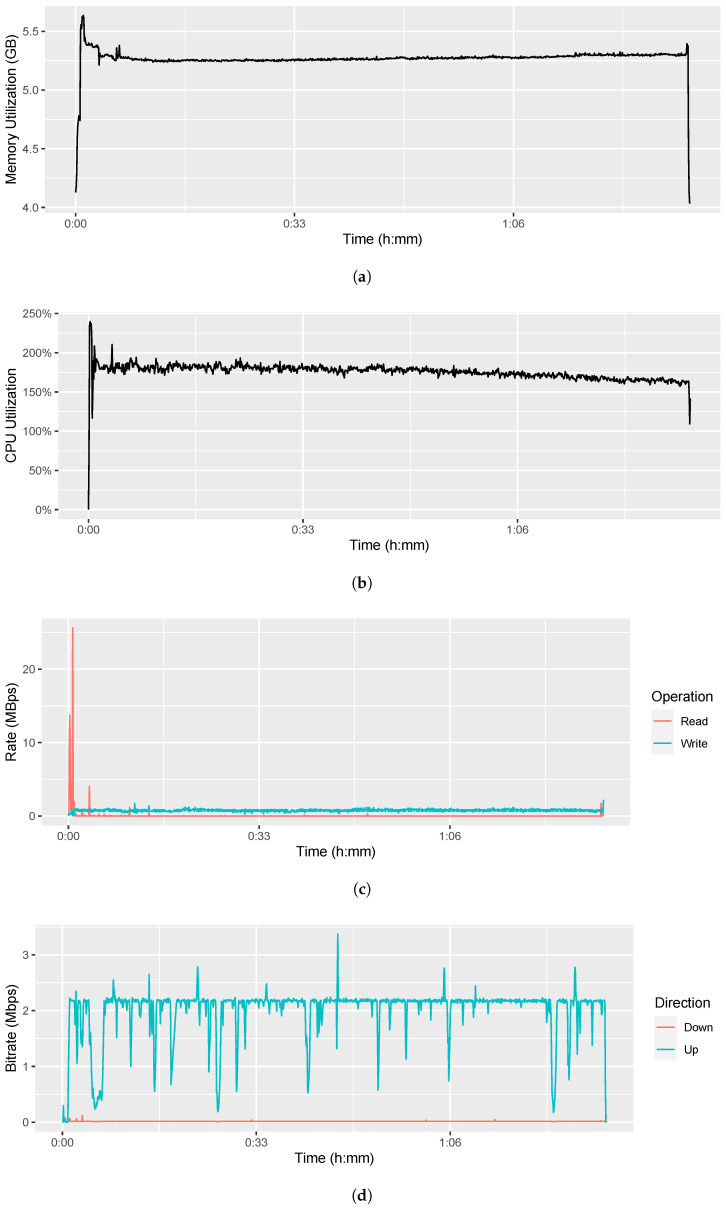
Performance evaluation of the Aggregator Device. (**a**) Memory utilization. (**b**) CPU utilization. (**c**) I/O disk utilization. (**d**) Network utilization.

**Figure 24 sensors-22-09589-f024:**
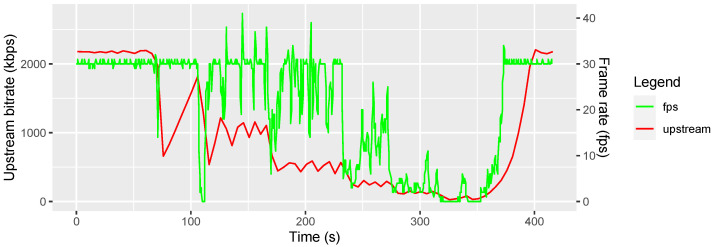
Frame rate in frames per second (fps) over time depending on a variable upstream bandwidth.

**Figure 25 sensors-22-09589-f025:**
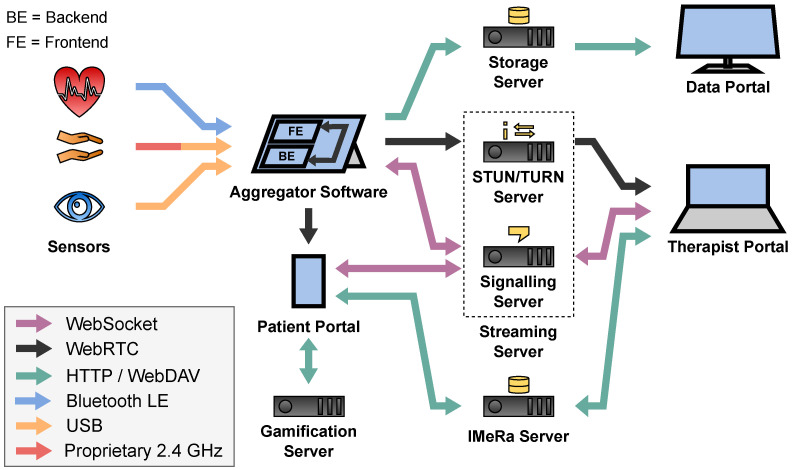
Overview of communication technologies among components in the SKS.

**Table 1 sensors-22-09589-t001:** Overview of sensor specifications.

	Sensor	Type	Sample Rate	Resolution	Connectivity
eye tracker	Left Eye	video	30 Hz	640 × 480 b/w	USB 2.0
	Right Eye		30 Hz	640 × 480 b/w	
	Field		30 Hz	640 × 480 color	
APDM Opal	Acceleration	3-axes	20–128 Hz	±16 g–±200 g	Proprietary 2.4 GHz and USB 2.0
	Angular Velocity		20–128 Hz	±2000 deg/s	
	Magnetic Field		20–128 Hz	±8 Gauss	
Movesense HR2	ECG	1-channel	125–512 Hz		Bluetooth 4.0/5.0
	Acceleration	3-axes	12.5–208 Hz	±2 g–±16 g	
	Angular Velocity		12.5–208 Hz	±2000 deg/s	
	Magentic Field		12.5–208 Hz	±49 gauss	

**Table 2 sensors-22-09589-t002:** Overview of servers, their location, and their provided services.

Server	Location	Services
IMeRa	Medical Datacenter Tübingen	IMeRa Server and DB
Keycloak	Medical Datacenter Tübingen	Keycloak Server and DB
Nextcloud	Datacenter Frankfurt	Nextcloud Server and DB, coturn
Services	University Datacenter Tübingen	Traefik, Coturn, Signalling, Gamification Server, and Admin, Therapist, and Patient Portal’s Nginx

**Table 3 sensors-22-09589-t003:** Storage requirements over time per sensor.

Sensor	Requirements	Percentage
Eye Tracking (video)	2637.5 MBph	97.3%
Eye Tracking (gaze estimation)	24.0 MBph	0.8%
APDM Movement	29.4 MBph	1.1%
Movesense Movement	1.5 MBph	0.5%
Movesense ECG	17.7 MBph	0.6%
Total	2710.1 MBph	100%

**Table 4 sensors-22-09589-t004:** Average CPU utilization of the Aggregator Device by process.

Application	CPU Utilization
Look Eye Tracking	56.83%
Aggregator Frontend	9.39%
Aggregator Backend	29.98%
Others	2.11%
Windows Defender	1.02%
Windows System	0.67%

## Data Availability

Not applicable.

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
