# Peer review of "The SSTeP-KiZ System—Secure Real-Time Communication Based on Open Web Standards for Multimodal Sensor-Assisted Tele-Psychotherapy"

_sensors, 2022, doi:10.3390/s22249589_

Round 1
Reviewer 1 Report
Dear Authors
the article is very valuable and congratulations on writing it. However, it seems that its layout does not correspond to the standards adopted in medical journals. Part of the material and methods is missing, in which the Authors should clearly write what material (or data) they used and what methods they used for analysis or description. As I understand it, the analyzed materials are the technical specifications of the devices, and the method of analysis is the description of how they are connected and how they work. This paragraph is very necessary for a better understanding of what we are writing about. In addition, medical science references are generally required in the VANCUVER style (end indexes) and not as subscripts. I am not able to fully understand how the eyeball tracking sensor is placed on the patient's body (photo in Figure 15 suggests a sensor on the wrist - the sensor with this position allows you to collect data about the patient's activity (patient movement) and basic parameters (temperature, ECG, etc.) ) on the other hand, it is not possible to track the movement of the eyeballs, does it mean that this aspect is realized by a webcam at, for example, an aggregator?
Dear authors, you are talking about treatment, but there are practically no medical components in the article! So: what diseases and what specialists will benefit from it! are they rehabilitators or psychologists? In each of these cases (rehabilitation vs. psychologist), other parameters are needed in the observation, and these need to be refined (e.g. a psychologist will be able to follow the eye movement better than a physiotherapist).
Author Response
Dear Reviewer 1, thank you very much for your feedback. Please see the full reply attatched.

Reviewer 2 Report
This paper is introducing the full software and hardware description of IoMT system for tele-psychotherapy. The authors used open standards to describe the system infrastructure. The system is applied in an ongoing study (SSTeP-KiZ) on the authors’ side. The authors combine video therapy with wearable sensors to deliver exposure and response prevention therapy aimed at training the avoidance of compulsive behaviors in daily home life. Also, the authors mentioned that a secure implementation of data storing, and streaming has been realized. After reading this paper, I have the following comments:
Major Comments:
- The main contribution in this paper, as mentioned by the authors is combining video therapy with wearable sensors to deliver exposure and response prevention therapy aimed at training the avoidance of compulsive behaviors. Also, the security of data storage and streaming has been realized. But I see that there are so many missing issues.
- The detailed specifications and description of the used sensors are missing.
- Using BitLocker is not enough to secure the stored data for unauthorized access.
- The access control details are missing. Thus, how would you allow multiple/different therapists to access the data?
- The schematic diagram which shows the connections among the devices is missing.
- The sensors’ calibration is missing.
- Figure 9 is redundant.
- The results focus on memory utilization, CPU, disk rate, and bit rate. But these are not the big issues for the system. You should care about the video processing performance and the performance of your system to realize the movement characteristics. They are missing in this paper.
- Listing multiple references as a range is not so acceptable. For example [1-6] and [10-14]. Especially because there is a huge variance in years among them. It is better to set one statement for each reference and you should consider sorting them based on the year of publication.
Minor Comments:
- There are some typos and grammatical issues in this paper.
- The paragraphs’ size is inconsistent.
Author Response
Dear Reviewer 2, thank you very much for your valuable feedback. Please see the full reply attached.

Reviewer 3 Report
The paper is very well organized.
However, the proposed technique requires discussion and focus with justification.
Author Response
Dear Reviewer 3, thank you very much for your feedback. Please see the full reply attached.
